# Benzophenone and Benzoylphloroglucinol Derivatives from *Hypericum sampsonii* with Anti-Inflammatory Mechanism of Otogirinin A

**DOI:** 10.3390/molecules25194463

**Published:** 2020-09-28

**Authors:** Chun-Yi Huang, Tzu-Cheng Chang, Yu-Jing Wu, Yun Chen, Jih-Jung Chen

**Affiliations:** 1School of Pharmaceutical Sciences, Faculty of Pharmacy, National Yang-Ming University, Taipei 11221, Taiwan; jimmyhuang9289@gmail.com (C.-Y.H.); joy0710082035@gmail.com (Y.C.); 2Department of Forestry and Natural Resources, National Ilan University, Yilan City 26047, Taiwan; tcchang@ems.niu.edu.tw; 3Institute of Pharmacology, National Yang-Ming University, Taipei 11221, Taiwan; kittylv199315@gmail.com; 4Department of Medical Research, China Medical University Hospital, China Medical University, Taichung 404332, Taiwan

**Keywords:** *Hypericum sampsonii*, Hypericaceae, structure elucidation, benzophenone, benzoylphloroglucinol derivative, anti-inflammatory activity

## Abstract

Three new compounds, 4-geranyloxy-2-hydroxy-6-isoprenyloxybenzophenone (**1**), hypericumone A (**2**) and hypericumone B (**3**), were obtained from the aerial parts of *Hypericum sampsonii*, along with six known compounds (**4**–**9**). The structures of these compounds were determined through spectroscopic and MS analyses. Hypericumone A (**2**), sampsonione J (**8**) and otogirinin A (**9**) exhibited potent inhibition (IC_50_ values ≤ 40.32 μM) against lipopolysaccharide (LPS)-induced nitric oxide (NO) generation. Otogirinin A (**9**) possessed the highest inhibitory effect on *NO* production with IC_50_ value of 32.87 ± 1.60 μM. The well-known proinflammatory cytokine, tumor necrosis factor-alpha (TNF-α) was also inhibited by otogirinin A (**9**). Western blot results demonstrated that otogirinin A (**9**) downregulated the high expression of inducible nitric oxide synthase (iNOS). Further investigations on the mechanism showed that otogirinin A (**9**) blocked the phosphorylation of MAPK/JNK and IκBα, whereas it showed no effect on the phosphorylation of MAPKs/ERK and p38. In addition, otogirinin A (**9**) stimulated anti-inflammatory M2 phenotype by elevating the expression of arginase 1 and Krüppel-like factor 4 (KLF4). The above results suggested that otogirinin A (**9**) could be considered as potential compound for further development of NO production-targeted anti-inflammatory agent.

## 1. Introduction

*Hypericum sampsonii* Hance (Hypericaceae) has been used as a traditional medicine herb for reducing blood stasis, relieving swelling and detoxification in Taiwan [1,2]. Diverse xanthones [3,4,5], benzophenones [6,7,8], bisanthraquinones [9], flavonoids [9,10] and polyprenylated phloroglucinols [1,11,12,13,14,15], and their derivatives have been isolated from this species in the past studies. Many of these isolated compounds show cytotoxic [3,14], anti-HIV [13], antibacterial [4,8], antitumor [10] and anti-inflammatory [10] activities. Abnormal inflammatory response causes a variety of diseases including asthma, Alzheimer’s disease, rheumatoid arthritis and even cancer [16]. Macrophages are the primary immune cells which secrete nitric oxide (NO), a mediator in the inflammatory response involved in host defense [17]. Suppression of the abnormal activation of macrophages by drugs has been suggested as a way to ameliorate inflammatory diseases. In our studies on the anti-inflammatory constituents of Chinese herbal medicines, many species have been screened for anti-inflammatory effect, and *H. sampsonii* was found to be an active species. Sampsoninone J (**8**) and otogirinin A (**9**) belong to the polyprenylated benzoylphloroglucinol derivatives with an unusual adamantyl skeleton. According to past studies, there was no biologic activity report for **8** and **9**, except that **8** showed no significant cytotoxicity against P338 cell line [18]. However, their analogous benzoylphloroglucinol derivatives, garcimultiflorone G [19] and sampsonione B [20] was reported to exhibit anti-inflammatory activity. This report depicts the structural elucidation of three new Compounds **1**–**3**, the inhibitory activities of all isolated compounds against LPS-induced NO generation and the anti-inflammatory mechanism of otogirinin A (**9**).

## 2. Results and Discussion

### 2.1. Effect of Different Fractions of MeOH Extract and Isolation of Compounds

The *n*-hexane fraction of the MeOH extract of *H. sampsonii* showed the most potent inhibition against lipopolysaccharide (LPS)-induced nitric oxide (NO) accumulation in RAW264.7 macrophages, while H_2_O fraction had no effect on NO generation (Figure 1). The murine macrophage cell line, RAW264.7, is often used to initially screen natural products for anti-inflammatory activity [21,22]. Based on this reason, we carried out the chromatographic purification of the hexane-soluble fraction of MeOH extracts of aerial parts of *H. sampsonii* on a silica gel column and preparative TLC afforded three undescribed (**1–3**) and six known compounds (**4**–**9**) (Figure 2).

Compound **1** was isolated as yellowish oil. Its molecular formula, C_28_H_34_O_4_, was determined on the basis of the negative HR–ESI–MS ion at *m*/*z* 433.2370 [M − H]^−^ (calcd 433.2379) and supported by the ^1^H- and ^13^C-NMR data. The presence of conjugated carbonyl group was revealed by the band at 1618 cm^−1^ in the IR spectrum and was confirmed by signal at δ 199.6 in the ^13^C-NMR spectrum. The ^1^H- and ^13^C-NMR data of **1** showed the presence of a benzoyl group [δ_H_ 7.35 (2H, br t, *J* = 7.5 Hz, H-10 and H-12), 7.43 (1H, br t, *J* = 7.5 Hz, H-11) and 7.48 (2H, br d, *J* = 7.5 Hz, H-9 and H-13); δ_C_ 127.4 (C-9), 127.4 (C-13), 127.5 (C-10), 127.5 (C-12), 130.2 (C-11) and 142.3 (C-8)], a hydroxyl group [δ_H_ 12.37(1H, s, D_2_O exchangeable, OH-2)], a geranyloxy group [δ_H_ 1.62, 1.69 and 1.76 (each 3H, each s, H-10ʹ, H-8ʹ and H-9ʹ), 2.11 (2H, br t, *J* = 6.5 Hz, H-4ʹ), 2.14 (2H, m, H-5ʹ), 4.58 (2H, d, *J* = 6.5 Hz, H-1ʹ), 5.10 (1H, br t, *J* = 6.5 Hz, H-6ʹ) and 5.48 (1H, br t, *J* = 6.5 Hz, H-2ʹ); δ_C_ 16.7 (C-9ʹ), 17.7 (C-10ʹ), 25.7 (C-8ʹ), 26.3 (C-5ʹ), 39.5 (C-4ʹ), 65.2 (C-1ʹ), 118.4 (C-2ʹ), 123.6 (C-6ʹ), 131.9 (C-7ʹ) and 142.3 (C-3ʹ)], an isoprenyloxy group [δ_H_ 1.50 and 1.57 (each 3H, each s, H-5ʹʹ and H-4ʹʹ), 4.18 (2H, d, *J* = 6.0 Hz, H-1ʹʹ) and 4.62 (1H, br t, *J* = 6.0 Hz, H-2ʹʹ); δ_C_ 18.0 (C-5ʹʹ), 25.5 (C-4ʹʹ), 65.1 (C-1ʹʹ), 118.4 (C-2ʹʹ) and 137.0 (C-3ʹʹ)] and two *meta*-coupling aromatic protons [δ_H_ 5.93 and 6.16 (each 1H, each d, *J* = 2.5 Hz, H-5 and H-3); δ_C_ 92.5 (C-5) and 94.1 (C-3)]. The signal at δ_H_ 12.37 exhibited a chelated hydroxyl group with the carbonyl group. These indicated there is a typical benzophenone structure in Compound **1** [15]. Comparison of the ^1^H- and ^13^C-NMR data of **1** with those of 4-geranyloxy-2,6-dihydroxybenzophenone [23] suggested that their structures were closely related, except that the 6-isoprenyloxy group of **1** replaced the 6-hydroxy group of 4-geranyloxy-2,6-dihydroxybenzophenone [23]. This was supported by both HMBC correlations between H-1ʹʹ (δ_H_ 4.18) and C-6 (δ_C_ 161.4), C-2ʹʹ (δ_C_ 118.4) and C-3ʹʹ (δ_C_ 137.0) and NOESY correlations between H-1ʹʹ (δ_H_ 4.18) and H-5 (δ_H_ 5.93). This was supported by ^1^H–^1^H COSY, HSQC, NOESY (Figure 3) and HMBC (Figure 3) experiments. On the basis of the above data, the structure of **1** was elucidated as 4-geranyloxy-2-hydroxy-6-isoprenyloxy- benzophenone.

Compound **2** was obtained as white amorphous powder, [α${]}_{D}^{25}$ = – 43.5 (*c*, 0.16, CHCl_3_). The ESI-MS demonstrated the quasi-molecular ion [M + Na]^+^ at *m*/*z* 511, implying a molecular formula of C_32_H_40_O_4_, which was confirmed by the HR–ESI–MS (*m*/*z* 511.2823 [M + Na]^+^, calcd 511.2824) and by the ^1^H- and ^13^C-NMR data. The presence of carbonyl groups was revealed by the bands at 1722 and 1684 cm^−1^ in the IR spectrum and was confirmed by the signals at δ 193.3, 211.6 and 212.3 in the ^13^C-NMR spectrum. The ^1^H- and ^13^C-NMR data of **2** showed the presence of a benzoyl group [δ_H_ 7.33 (2H, br dd, *J* = 8.4, 7.4 Hz, H-13 and H-15), 7.45 (1H, br t, *J* = 7.4 Hz, H-14) and 7.56 (2H, br d, *J* = 8.4 Hz, H-12 and H-16); δ_C_ 128.1 (C-13), 128.1 (C-15), 129.1 (C-12), 129.1 (C-16), 132.8 (C-14) and 134.5 (C-11)], a hydroxyl group [δ_H_ 3.38 (1H, s, D_2_O exchangeable, OH-3)], a geranyl group [δ_H_ 1.62, 1.68 and 1.69 (each 3H, each s, H-37, H-38 and H-32), 2.11 (2H, m, H-33), 2.14 (2H, m, H-34), 2.58 (2H, m, H-29), 5.10 (1H, br t, *J* = 6.7 Hz, H-35) and 5.40 (1H, br t, *J* = 7.5 Hz, H-30); δ_C_ 16.2 (C-32), 17.7 (C-37), 25.8 (C-38), 26.3 (C-29), 26.4 (C-34), 40.2 (C-33), 118.5 (C-30), 124.1 (C-35), 131.7 (C-36) and 139.0 (C-31)], a prop-1-en-2-yl group [δ_H_ 1.85 (3H, s, H-22), 5.01 (1H, br s, H-23) and 5.25 (1H, br s, H-23); δ_C_ 21.7 (C-22), 118.5 (C-23) and 142.5 (C-21)] and two methyl groups [δ_H_ 1.36, 1.44 (each 3H, each s, H-17 and H-18); δ_C_ 21.7 (C-17) and 24.5 (C-18)]. Comparison of the ^1^H- and ^13^C-NMR data of **2** with those of norsampsone E [24] suggested that their structures were closely related, except that the benzoyl group at C-1 of **2** replaced the 1-isobutyryl group of norsampsone E [24]. This was supported by both HMBC correlations between H-12 (δ_H_ 7.56) and C-10 (δ_C_ 193.3), C-14 (δ_C_ 132.8) and C-16 (δ_C_ 129.1) and ROESY correlations between H-12 (δ_H_ 7.56) and H-17 (δ_H_ 1.36). To further confirm the relative configuration of **2**, a computer-assisted 3D structure was obtained by using the molecular-modeling program CS CHEM 3D Ultra 12.0, with MM2 force-field calculations for energy minimization. The calculated distances between OH-3/H-23 (2.571 Å), H-6/H-18 (2.072 Å), H-6/H-20 (2.570 Å), H-6/H-29 (2.532 Å), H-12/H-17 (2.557 Å) are all less than 4 Å (Figure 4). This is consistent with the well-defined ROESY observed for each of these H-atom pairs. The ^1^H- and ^13^C-NMR resonances were fully assigned by ^1^H–^1^H COSY, HSQC, ROESY and HMBC experiments (Figure 5). On the basis of the above data, the structure of **2** was elucidated as shown in Figure 2 and named hypericumone A.

Compound **3** was isolated as colorless oil, [α${]}_{D}^{25}$ = + 326.4 (*c* 0.15, CHCl_3_). Its molecular formula, C_30_H_42_O_2_, was determined on the basis of the positive HR–ESI–MS (*m*/*z* 457.3086 [M + Na]^+^, calcd 457.3083) and supported by the ^1^H- and ^13^C-NMR data. The IR spectrum displayed the presence of carbonyl groups (1707 and 1672 cm^−1^), which was confirmed by the signals at δ 196.7 and 208.4 in the ^13^C-NMR spectrum. The ^1^H- and ^13^C-NMR data of **3** showed the presence of a benzoyl group [δ_H_ 7.45 (2H, br t, *J* = 7.5 Hz, H-3 and H-5), 7.55 (1H, br t, *J* = 7.5 Hz, H-4) and 7.99 (2H, br d, *J* = 7.5 Hz, H-2 and H-6); δ_C_ 128.6 (C-3), 128.6 (C-5), 128.7 (C-2), 128.7 (C-6), 133.2 (C-4) and 138.5 (C-1)], a geranyl group [δ_H_ 1.58, 1.59 and 1.66 (each 3H, each s, H-22, H-23 and H-21), 1.95 (2H, m, H-17), 1.98 (1H, m, H-14), 2.02 (2H, m, H-18), 2.32 (1H, m, H-14), 5.05 (1H, br t, *J* = 7.6 Hz, H-15) and 5.07 (1H, br t, *J* = 6.8 Hz, H-19); δ_C_ 16.2 (C-22), 17.7 (C-23), 25.7 (C-21), 26.8 (C-18), 27.4 (C-14), 39.8 (C-17), 121.6 (C-15), 124.3 (C-19), 131.3 (C-20) and 136.7 (C-16)], an isoprenyl group [δ_H_ 1.60 (3H, s, H-28), 1.67 (1H, m, H-24), 1.73 (3H, s, H-27), 2.15 (1H, m, H-24) and 5.21 (1H, br t, *J* = 7.4 Hz, H-25); δ_C_ 17.8 (C-28), 25.8 (C-27), 28.1 (C-24), 123.6 (C-25) and 132.4 (C-26)], two methyl groups [δ_H_ 0.91, 0.99 (each 3H, each s, H-29 and H-30); δ_C_ 22.1 (C-29) and 26.1 (C-30)] and a cyclohexanone core structure [δ_H_ 1.14 (1H, m, H_α_-11), 2.20 (1H, m, H_β_-11), 2.64 (1H, m, H-12), 2.88 (1H, m, H-10), 4.35 (1H, s, H-8); δ_C_ 34.5 (C-11), 40.3 (C-12), 43.3 (C-13), 47.6 (C-10), 72.1 (C-8) and 208.4 (C-9)]. Comparison of the ^1^H- and ^13^C-NMR data of **3** with those of norsampsone B [25] suggested that their structures were closely related, except that the 10-geranyl group and H-10 of **3** replaced the isoprenyl and 5-methylhex-4-enoyl groups at C-10 of norsampsone B [25]. This was supported by both HMBC correlations between H-14 (δ_H_ 1.98 and 2.32) and C-9 (δ_C_ 208.4), C-10 (δ_C_ 47.6) and C-11 (δ_C_ 34.5) and ROESY correlations between H-14 (δ_H_ 1.98 and 2.32) and H-10 (δ_H_ 2.88). The relative stereochemistry of **3** was elucidated on the basis of ROESY experiments (Figure 6). The ROESY cross-peaks between H-8/H-29, H-10/H-12, H_α_-11/H-14, H_α_-11/H-24, H_α_-11/H-29, H-12/H-30 and H-2/H-30 suggested that H-10, H-12, Me(30) and the bond between C-7 and C-8 is on the β-side and Me(29), H-8, the geranyl group at C-10 and the isoprenyl group at C-12 are on the α-side of **3**. A computer-assisted 3D structure was obtained by using the molecular modeling program CS CHEM 3D Ultra 12.0, with MM2 force-field calculations for energy minimization. The calculated distances between H-8/H-29 (2.768 Å), H-10/H-12 (2.507 Å), H_α_-11/H-14 (2.542 Å), H_α_-11/H-24 (3.247 Å), H_α_-11/H-29 (2.231 Å), H-12/H-30 (2.509 Å) and H-2/H-30 (3.631 Å) are all less than 4 Å (Figure 6); this is consistent with the well-defined ROESY observed for each of the proton pairs. Thus, the structure of **3** was elucidated as shown in Figure 2 and named hypericumone B. This was further confirmed by the ^1^H–^1^H COSY, ROESY, HSQC and HMBC experiments (Figure 7).

### 2.2. Structure Identification of the Known Isolates

The known isolates were readily identified by a comparison of physical and spectroscopic data (UV, IR, ^1^H-NMR, [α]_D_ and MS) with corresponding authentic samples or literature values, and this included two xanthones, 1-hydroxy-7-methoxyxanthone (**4**) [26] and 2-methoxyxanthone (**5**) [27], two benzophenones, 2,4,6-trihydroxybenzophenone-4-*O*-geranyl ether (**6**) [23] and sampsonin B (**7**) [7], two benzoylphloroglucinol derivatives, sampsonione J (**8**) [18] and otogirinin A (**9**) [28].

### 2.3. Biological Studies

#### 2.3.1. Reduction of LPS-Triggered Nitric Oxide (NO) Production in RAW264.7 Cells by Compounds **8** and **9**

Nitric oxide (NO) is a mediator in the inflammatory response involved in host defense. The anti-inflammatory effects of the compounds isolated from the aerial parts of *H. sampsonii* were evaluated by suppressing lipopolysaccharide (LPS)-induced NO generation in murine macrophage cell line RAW264.7. The inhibitory activity data of the isolated Compounds **1**–**9** on NO generation by macrophages are shown in Table 1. The anti-inflammatory agent, andrographolide (ANDRO), was used as positive control, which was reported to exhibit gastroprotective and ulcer preventive effects, along with its well-documented anti-inflammatory effects [29]. The results of anti-inflammatory assays show that sampsonione J (**8**) and otogirinin A (**9**) obviously suppressed NO production comparable with andrographolide, in a concentration-dependent manner in RAW264.7 macrophages (Figure 8A and Table 1). Compounds **8** and **9** are benzoylphloroglucinol derivatives with inhibition (%) values of 67.20 ± 3.29 and 73.10 ± 1.59%, respectively, against LPS-induced NO production at 50 μM. Otogirinin A (**9**) is the most effective among the isolated compounds, with IC_50_ = 32.87 ± 1.60 μM, against LPS-induced NO generation. Both of the Compounds **8** and **9** did not show significant cytotoxicity against RAW264.7 macrophages (Figure 8B), which suggested that inhibitory activities of Compounds **8** and **9** on LPS-induced NO production did not resulted from their cytotoxicities. Furthermore, the anti-inflammatory activity of otogirinin A (**9**) has never been reported previously, thus the detailed mechanism of action of Compound **9** seems to be worth of further studies.

#### 2.3.2. Effect of Otogirinin A (9) on Proinflammatory Mediators in LPS-Stimulated RAW264.7 Cells

Tumor necrosis factor-α (TNF-α), interleukin-6 (IL-6), inducible nitric oxide synthase (iNOS) and cyclooxygenase-2 (COX-2) are pivotal proinflammatory mediators during the inflammatory process [30]. We further investigated whether otogirinin A (**9**) was able to reduce these proinflammatory molecules induced by LPS. The results showed that otogirinin A (**9**) suppressed LPS-induced TNF-α generation but did not exhibit significant inhibitory activity against IL-6 production (Figure 9A). In addition, the expression level of iNOS was reduced by treatment with otogirinin A (**9**); however, there was no significant difference of COX-2 by treatment with otogirinin A (**9**) (Figure 9B). Therefore, otogirinin A (**9**) exhibited the anti-inflammatory activity by reducing iNOS protein expression instead of COX-2.

#### 2.3.3. Effect of Otogirinin A (9) on M2 Polarization of Macrophages in LPS-Stimulated RAW264.7 Cells

M2-polarized macrophages are important on tissue repair [31]. Arginase 1 is an important M2 marker that connects Krüppel-like factor 4 (KLF4) to the biologic processes involved in M2 polarization [32]. High level of arginase-1 can compete with iNOS for arginine and reduce NO production [33]. In addition, KLF4, which is one of the major members of the KLF family, was shown to induce M2 macrophage phenotype, whereas it reduced M1 macrophage expression [34]. We further examined whether otogirinin A (**9**) enhanced the expression level of M2 macrophages. The result showed that expression levels of arginase-1 and KLF4 were both induced by treatment with otogirinin A (**9**) with a concentration-dependent manner (Figure 10A,B). These results suggested that otogirinin A (**9**) promoted macrophage M2 markers, arginase-1 and KLF4, which exhibited the anti-inflammatory activity.

#### 2.3.4. Effect of Otogirinin A (**9**) on MAPK Phosphorylation Pathway in LPS-Stimulated RAW264.7 Cells

Mitogen-activated protein kinases (MAPKs) are crucial transcription factors for anti-inflammatory therapeutics because their activation and involvement in regulation of the synthesis of inflammation mediators at transcriptional and translational levels [35]. To investigate the MAPK pathway, we examined whether otogirinin A (**9**) effected on LPS-induced phosphorylation of extracellular signal-regulated kinase (ERK), c-JUN N-terminal kinase (JNK) and p38 MAPKs by Western blot analysis. The result showed that the expression levels of pJNK were dramatically reduced by treatment with otogirinin A (**9**) (Figure 11A). However, no significant difference of pp38 and pERK was observed when treated with otogirinin A (**9**) (Figure 11B,C). Therefore, otogirinin A (**9**) suppressed MAPK/JNK signaling pathway by reducing the phosphorylation of JNK instead of p38 and ERK.

#### 2.3.5. Effect of Otogirinin A (**9**) on IκBα Phosphorylation in LPS-Stimulated RAW264.7 Cells

NF-κB is a potential target in the inflammation process and known to be a molecular target of anti-inflammatory drugs. Upon LPS activation, IκB kinase (IKK) phosphorylates IκBα and triggers ubiquitin-dependent IκBα degradation in the proteasome, resulting in rapid and transient nuclear translocation of NF-κB [36]. Therefore, we examined if otogirinin A (**9**) could suppress the degradation of IκBα and translocation of NF-κB into the nucleus. The result indicated that otogirinin A (**9**) elevated the protein expression of IκBα, which attenuated IκBα degradation and thus inhibited the translocation of NF-κB into the nucleus in LPS-stimulated RAW264.7 macrophages (Figure 11D). These results reveal that the suppression of degradation of IκBα by otogirinin A (**9**) may be involved in the inhibition of LPS-stimulated NF-κB activation in RAW264.7 cells.

## 3. Experimental Section

### 3.1. General Procedures

Optical rotations were measured using a Jasco *P*-2000 polarimeter (Japan Spectroscopic Corporation, Tokyo, Japan) in CHCl_3_. Ultraviolet (UV) spectra were obtained on a Hitachi *U*-2800 Double Beam Spectrophotometer (Hitachi High-Technologies Corporation, Tokyo, Japan). Infrared (IR) spectra (neat or KBr) were recorded on a Shimadzu IRAffinity-1S FT-IR spectrophotometer (Shimadzu Corporation, Kyoto, Japan). Nuclear magnetic resonance (NMR) spectra, including correlation spectroscopy (COSY), nuclear Overhauser effect spectrometry (NOESY), rotating frame nuclear Overhauser effect spectrometry (ROESY), heteronuclear multiple-bond correlation (HMBC) and heteronuclear single-quantum coherence (HSQC) experiments, were acquired using a BRUKER AVIII-500 or a BRUKER AVIII-600 spectrometer (Bruker, Bremen, Germany) operating at 500 or 600 MHz (^1^H) and 125 or 150 MHz (^13^C), respectively, with chemical shifts given in ppm (δ) using CDCl_3_ as an internal standard (peak at 7.263 ppm). HPLC separations were performed utilizing a LC-2000 Plus HPLC system (Jasco, Tokyo, Japan) equipped with PU-2080 Plus pumps and a MD-2010 Plus diode array detector, using ChromNav software (version 2.0, Jasco). Cosmosil 5SL-II (Nacalai Tesque, 5 μm; columns of dimensions 10 × 250 mm) columns were used for semipreparative HPLC. Electrospray ionization (ESI) and high-resolution electrospray ionization (HRESI)-mass spectra were recorded on a Bruker APEX II mass spectrometer (Bruker, Bremen, Germany) or a (HPLC/MS-MS) VARIAN and VARIAN 901-MS (Varian, Inc., Palo Alto, CA, USA). Silica gel (70–230, 230–400 mesh, Merck) was used for column chromatography (CC). Silica gel 60 F-254 (Merck, Darmstadt, Germany) was used for thin-layer chromatography (TLC) and preparative thin-layer chromatography (PTLC).

### 3.2. Plant Material

The aerial parts of *Hypericum sampsonii* Hance (Hyperiaceae), bought from Li-Fa pharmaceutical company, were collected from Guangxi, China, in September 2016 and identified by Prof. J.-J. Chen. A voucher specimen (HS-201,609) was deposited in the Faculty of Pharmacy, National Yang-Ming University, Taipei, Taiwan.

### 3.3. Extraction and Isolation

The dried aerial parts (3.1 kg) of *H. sampsonii* were extracted three times with MeOH (15 L each) for three days. The extract was concentrated under reduced pressure at 35 °C, and the residue (192.4 g) was partitioned between *n*-hexane and H_2_O (1:1) to provide the *n*-hexane-soluble fraction (fraction A; 82.1 g). The H_2_O-soluble fraction was further extracted with CH_2_Cl_2_ and provided CH_2_Cl_2_-soluble part (fraction B; 17.9 g). Finally, the H_2_O-soluble one was extracted with EtOAc, and the EtOAc-soluble part (fraction C; 16.1 g) and the H_2_O-soluble one (fraction D; 62.9 g) were separated. Fraction A (82.1 g) was purified by CC (4.0 kg of silica gel, 63–200 mesh; *n*-hexane/EtOAc gradient) to afford 11 fractions: A1–A11. Fraction A1 (1.44 g) was subjected to CC (60 g of silica gel, 230–400 mesh; *n*-hexane/EtOAc 1:0–9:1, 500 mL-fractions) to give 11 subfractions: A1-1–A1-11. Fraction A1-8 (302 mg) was purified by preparative TLC (silica gel; *n*-hexane/CH_2_Cl_2_ 3:2) and collected the major part. Then the major part was further purified by preparative TLC (silica gel; *n*-hexane/EtOAc 9:1) to obtain 1-hydroxy-7-methoxyxanthone (**4**) (8.5 mg) and sampsonin B (**7**) (2.7 mg). Fraction A2 (5.88 g) was subjected to CC (250 g of silica gel, 230–400 mesh; *n*-hexane/EtOAc 1:0–9:1, 500 mL-fractions) to give 10 subfractions: A2-1–A2-10. Part (117 mg) of fraction A2-1 was purified by preparative TLC (silica gel; *n*-hexane/a etone 19:1) and collected the major part. Then the major part was further purified by silica HPLC (Cosmosil 5SL-II column, 10 × 250 mm) with *n*-hexane/CH_2_Cl_2_ (4:6, 2.0 mL/min) to yield hypericumone A (**2**) (2.5 mg, eluted at 11.25 min). Part (133 mg) of fraction A2-4 was further purified by preparative TLC (silica gel; *n*-hexane/CH_2_Cl_2_ 4:6) to afford 2-methoxyxanthone (**5**) (3.1 mg). Part (122 mg) of fraction A2-8 was further purified by preparative TLC (silica gel; *n*-hexane/CH_2_Cl_2_ 1:1) to afford otogirinin A (**9**) (9.3 mg). Fraction A3 (5.96 g) was subjected to CC (250 g of silica gel, 230–400 mesh; *n*-hexane/EtOAc 1:0–9:1, 500 mL-fractions) to give 10 subfractions: A3-1–A3-10. Part (115 mg) of fraction A3-2 was purified by preparative TLC (silica gel; *n*-hexane/CH_2_Cl_2_ 3:7) and collected the major part. Then the major part was further purified by preparative TLC again (silica gel; *n*-hexane/EtOAc 6:1) to obtain sampsonione J (**8**) (1.2 mg). Part (129 mg) of Fraction A3-4 was further purified by preparative TLC (silica gel; *n*-hexane/EtOAc 6:1) to afford 4-geranyloxy-2-hydroxy-6-isoprenyloxybenzophenone (**1**) (4.8 mg). Fraction A5 (1.75 g) was subjected to CC (80 g of silica gel, 230–400 mesh; *n*-hexane/EtOAc 1:0–9:1, 500 mL-fractions) to give 8 subfractions: A5-1–A5-8. Part (131 mg) of fraction A5-4 was purified by preparative TLC (silica gel; *n*-hexane/CH_2_Cl_2_ 1:1) and collected the major part. Then the major part was purified by silica HPLC (Cosmosil 5SL-II column, 10 × 250 mm) with *n*-hexane/CH_2_Cl_2_ (7:3, 2.0 mL/min) to yield hyperhexanone C (**3**) (2.3 mg, eluted at 22.33 min). Fraction A6 (2.61 g) was subjected to CC (150 g of silica gel, 230–400 mesh; *n*-hexane/EtOAc 1:0–9:1, 500 mL-fractions) to give 10 subfractions: A6-1–A6-10. Part (117 mg) of fraction A6-2 was further purified by preparative TLC (silica gel; *n*-hexane/CH_2_Cl_2_ 3:7) to afford 2,4,6-trihydroxybenzophenone-4-*O*-geranyl ether (**6**) (7.2 mg).

4-Geranyloxy-2-hydroxy-6-isoprenyloxybenzophenone (**1**): Yellowish oil; UV (MeOH): λ_max_ (log ε) = 304 (4.12) nm; IR: υ_max_ = 3428 (OH), 2922, 1618 (C=O), 1578, 1323, 1279, 1211, 1152, 1099 cm^−1^; ^1^H-NMR spectroscopic data, see Table 2; ^13^C-NMR spectroscopic data, see Table 3; HR–ESI–MS: *m*/*z* = 433.2370 [M − H]^–^ (calcd for C_28_H_33_O_4_, 433.2379); HR-MS, 1D-, and 2D-NMR spectra, see Appendix A. 

Hypericumone A (**2**): white amorphous powder; [α${]}_{D}^{25}$= –43.5˚ (*c* 0.16, CHCl_3_); UV (MeOH): λ_max_ (log ε) = 249 (4.15) nm; IR: υ_max_ = 3495 (OH), 2926, 1722 (C=O), 1684 (C=O), 1446, 1259, 1109 cm^−1^; ^1^H-NMR spectroscopic data, see Table 2; ^13^C-NMR spectroscopic data, see Table 3; HR–ESI–MS: *m*/*z* = 511.2824 [M + Na]^+^ (calcd for C_32_H_40_O_4_Na, 511.2824); HR-MS, 1D-, and 2D-NMR spectra, see Appendix A.

Hypericumone B (**3**): colorless oil; [α${]}_{D}^{25}$ = + 326.4˚ (*c* 0.15, CHCl_3_); UV (MeOH): λ_max_ (log ε) = 249 (4.10) nm; IR: υ_max_ = 1707 (C=O), 1672 (C=O), 1446, 1211, 1184, 1080 cm^−1^; ^1^H-NMR spectroscopic data, see Table 2; ^13^C-NMR spectroscopic data, see Table 3; HR–ESI–MS: *m*/*z* = 457.3086 [M + Na]^+^ (calcd for C_30_H_42_O_2_Na, 457.3083); HR-MS, 1D-, and 2D-NMR spectra, see Appendix A.

### 3.4. Biologic Assay

The effect of the isolated compounds on macrophage proinflammatory response was evaluated by monitoring the inhibition of nitric oxide (NO), iNOS, TNF-α and IL-6 generation. The purity of the tested compounds was >98% as identified by NMR and MS.

#### 3.4.1. Chemicals and Antibodies

Andrographolide (ANDRO), LPS and bovine serum albumin (BSA) were purchased from Sigma-Aldrich (St. Louis, MO, USA). The antibodies against iNOS, IκB-α, ERK, phospho-ERK (Thr202/Thr204), JNK, phospho-JNK (Thr183/Tyr185), p38, phospho-p38 (Thr180/Tyr182), KLF4, arginase 1 and glyceraldehyde-3-phosphate dehydrogenase (GAPDH) were purchased from Cell Signaling Technology (Danvers, MA, USA). Anti-COX-2 antibody was purchased from Arigo (Hsinchu, Taiwan). Anti-β-actin antibody was purchased from Sigma-Aldrich.

#### 3.4.2. Cells and Culture Medium

Preparation of RAW264.7 murine macrophages were carried out in accord with methods discussed in the literatures [37]. The murine macrophage cell line RAW264.7 was cultures in DMEM supplemented with 10% heat-inactivated fetal bovine serum (FBS), 100 U/mL penicillin and 100 µg/mL streptomycin. Cells were incubated at 37 °C humidified 5% CO_2_ atmosphere in a fully humid atmosphere.

#### 3.4.3. Determination of NO Production

The murine macrophage cell line RAW264.7 (4 × 10^5^ cells in 96-well plates) was cultured in DMEM supplemented with 10% heat-inactivated fetal bovine serum (FBS) and incubated at 37 °C humidified 5% CO_2_ atmosphere with a 96-well flat-bottomed culture plate. After 24 h, RAW264.7 cells were pretreated with Compounds **1**‒**9** (0, 6.25, 12.5, 25 and 50 μM) or vehicle (0.1% DMSO) for 1 h, respectively and incubated in the presence of LPS (100 ng/mL) under the same condition for 20 h. The cultured cells were then centrifuged and the supernatants were used for NO-production measurement. The supernatant was mixed with an equal volume of the Griess reagent (1% sulfanilamide, 0.1% *N*-(naphthalen-1-yl)ethylenediamine dihydrochloride in 2.5% H_2_PO_4_ soln.) and incubated for 15 min at room temperature. Nitrite concentration was determined by measuring the absorbance at 550 nm using an ELISA plate reader (μ Quant) [38].

#### 3.4.4. MTT Assay

A MTT colorimetric assay was used to determine cell viability. The assay was modified from that of Mosmann [39]. MTT reagent (0.5 mg/mL) was added onto the attached cells mentioned above (100 μL per 100 μL culture) and incubated at 37 °C for 3 h. Then, DMSO was added and amount of colored formazan metabolite formed was measured by absorbance at 570 nm using an ELISA plate reader (μ Quant). The optical density of formazan formed in control (untreated) cells was taken as 100% viability.

#### 3.4.5. Enzyme-Linked Immunosorbent Assay

RAW264.7 cells (4 × 10^5^ cells in 96-well plates) were pretreated with Compound **9** or vehicle (0.1% DMSO) for 1 h and then incubated with LPS (100 ng/mL) for 20 h. Supernatants were collected and analyzed for production of TNF-α and IL-6 by using appropriate ELISA kits (R&D, MN, USA) in accordance with the manufacturer’s instructions.

#### 3.4.6. Western Blotting

Western blot analysis followed as previously described with slight changes [40]. Then, 1.0 × 10^6^ cells were seeded into 6 cm dish and grown until 80–85% confluent. Compound **9** and LPS (100 ng/mL) were added or were not added. After 15 min (for detecting p-IκB, p-ERK, p-JNK and p-p38) and 24 h (for detecting iNOS, COX-2, arginase 1 and KLF4) of incubation at 37 °C, cultured medium was removed, and cells were washed with ice-cold PBS. After RIPA buffer (Cell Signaling, MA, USA) was added, cells were scraped off the plate and transferred to the Eppendorf on ice immediately. The proteins were quantified using the BCA protein assay. Cells were preserved at ‒ 80 °C overnight and then centrifuged (15,000 × rpm, 30 min, 4 °C). Equal amount of protein samples (25 μg) and pre-stained protein marker were loaded onto SDS-PAGE. After being stacked at 80 V and separated at 100 V, the proteins were transferred onto the polyvinylidene fluoride (PVDF) membranes at 350 mA. The PVDF membranes were blocked with 5% (*w*/*v*) of BSA with Tris-buffered saline (TBST) containing 0.1% (*v*/*v*) Tween-20 at room temperature for 1 h and washed three times with TBST for 15 min each time. Primary antibodies were incubated with the membranes overnight, shaking at 4 °C. Then each membrane was washed with TBST and incubated with horseradish peroxidase (HRP)-conjugated secondary antibodies at room temperature for 1 h while shaking. Finally, each membrane was developed using ECL detection kit, and the images were visualized by ImageQuant LAS 4000 mini (GE Healthcare, MA, USA). Band densities were quantified using ImageJ software (BioTechniques, NY, USA).

#### 3.4.7. Statistical Analysis

All data are expressed as mean ± SEM Statistical analysis was carried out using Student’s *t*-test. A probability of 0.05 or less was considered statistically significant. Microsoft Excel 2010 was used for the statistical and graphic evaluation. All the experiments were performed at least three times.

## 4. Conclusions

Nine isolates, including three new compounds, 4-geranyloxy-2-hydroxy-6-isoprenyl-oxybenzo- phenone (**1**), hypericumone A (**2**) and hypericumone B (**3**), were isolated from the aerial parts of *H. sampsonii*. Structures of these compounds were established by spectroscopic data. Hypericumone A (**2**), sampsonione J (**8**) and otogirinin A (**9**) exhibited potent inhibition (IC_50_ values ≤ 40.32 μM) against LPS-induced nitric oxide (NO) generation. Otogirinin A (**9**) was the most effective among the isolated compounds, with IC_50_ value of 32.87 ± 1.60 μM, against LPS-induced NO generation. Our results demonstrated that otogirinin A (**9**) suppressed LPS-induced iNOS expression and NO and TNF-α generation via blocking the phosphorylation of MAPK/JNK and degradation of IκBα, whereas it showed no effect on the phosphorylation of MAPKs/ERK and p38. In addition, otogirinin A (**9**) stimulated anti-inflammatory M2 phenotype by elevating the expression of arginase 1 and KLF4. In conclusion, otogirinin A (**9**) interfered with multiple intracellular targets. We can also draw a schematic diagram that how otogirinin A (**9**) influenced the polarization of M1 and M2 macrophages (Figure 12). The above results suggested that *H. sampsonii* and its isolates (especially **2**, **8** and **9**) are worth further research and may be expectantly developed as candidates for the treatment or prevention of various inflammatory diseases.

## Figures and Tables

**Figure 1 molecules-25-04463-f001:**
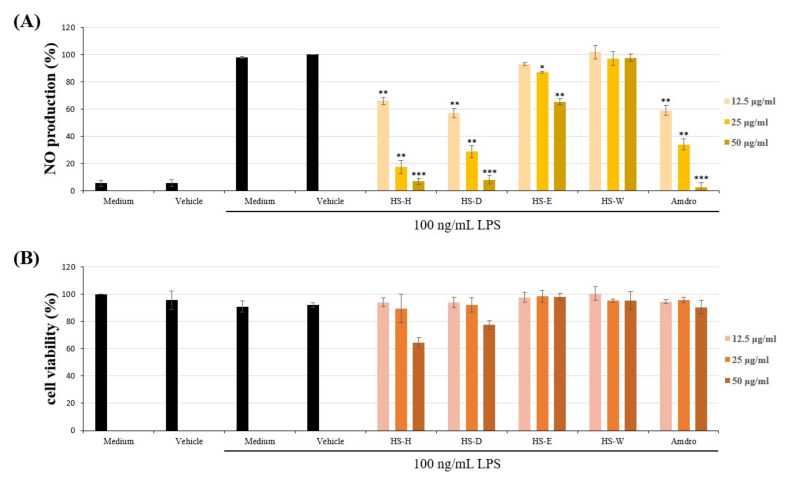
Effects of different partition fractions (HS-H, HS-D, HS-E and HS-W represent *n*-hexane, dichloromethane, ethyl acetate and H_2_O fractions, respectively) of crude MeOH extracts from *Hypericum sampsonii* on (**A**) nitric oxide (NO) production and (**B**) cytotoxicity in lipopolysaccharide (LPS)-induced RAW264.7 macrophages. Data expressed as means ± SEM *(n =* 3). Asterisks indicate significant differences (* *p* < 0.05, ** *p* < 0.01 and *** *p* < 0.001) from the (media + DMSO + LPS) control group.

**Figure 2 molecules-25-04463-f002:**
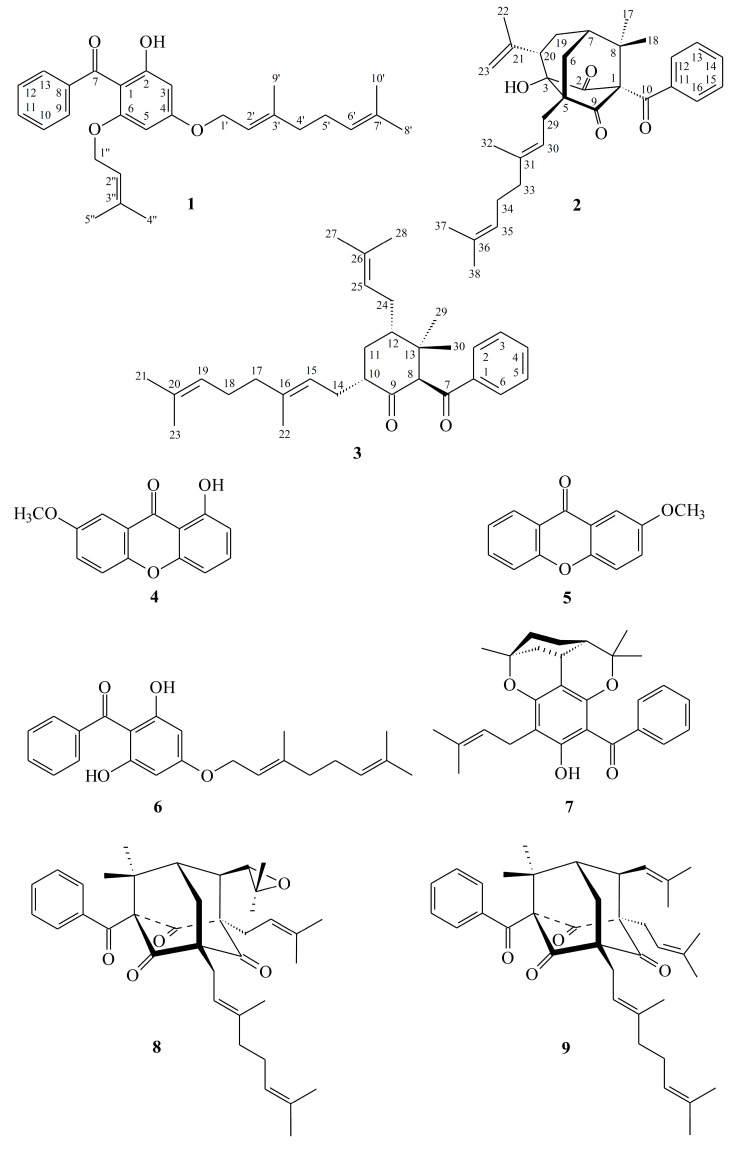
Chemical structures of Compounds **1**–**9** isolated from *H. sampsonii*.

**Figure 3 molecules-25-04463-f003:**
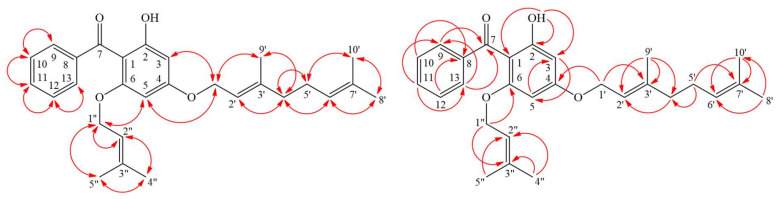
Key NOESY (
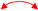
) and HMBC (
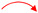
) correlations of **1**. NOESY—nuclear Overhauser effect spectrometry; HMBC—heteronuclear multiple-bond correlation.

**Figure 4 molecules-25-04463-f004:**
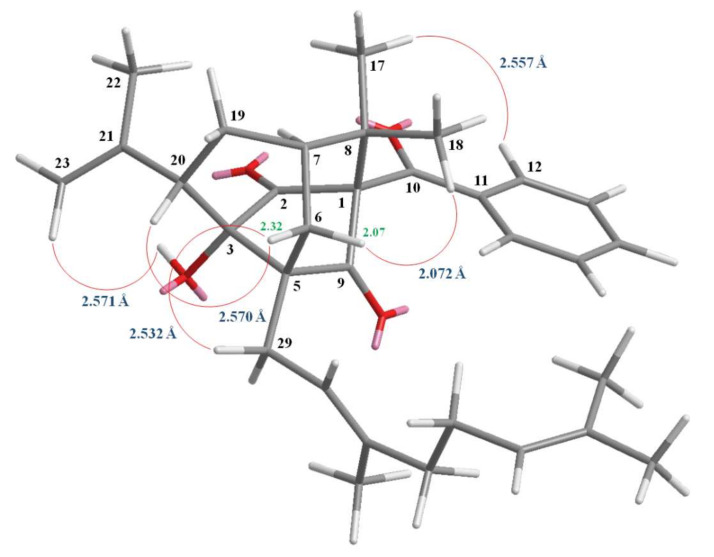
Selected rotating frame nuclear Overhauser effect spectrometry (ROESY) correlations and relative configuration of **2**.

**Figure 5 molecules-25-04463-f005:**
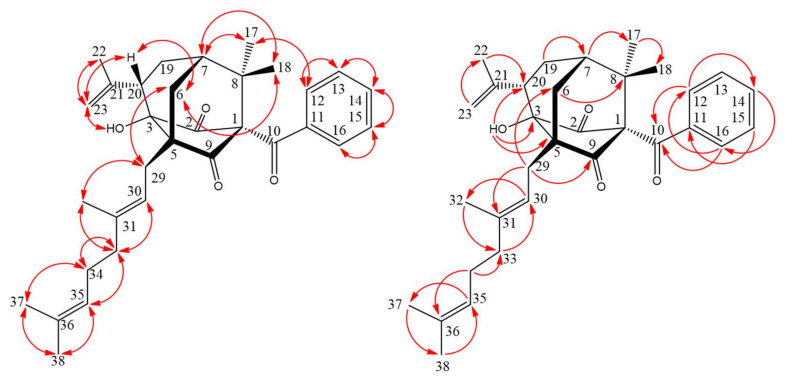
Key NOESY (
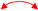
) and HMBC (
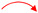
) correlations of **2**.

**Figure 6 molecules-25-04463-f006:**
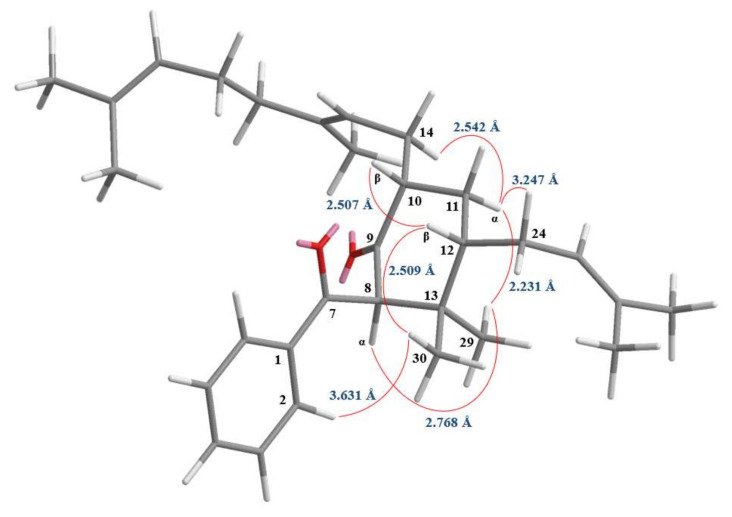
Selected ROESY correlations and relative configuration of **3**.

**Figure 7 molecules-25-04463-f007:**
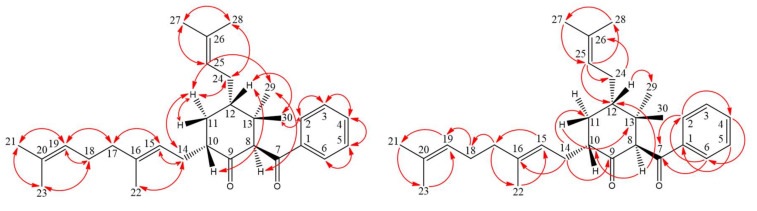
Key NOESY (
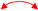
) and HMBC (
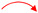
) correlations of **3**.

**Figure 8 molecules-25-04463-f008:**
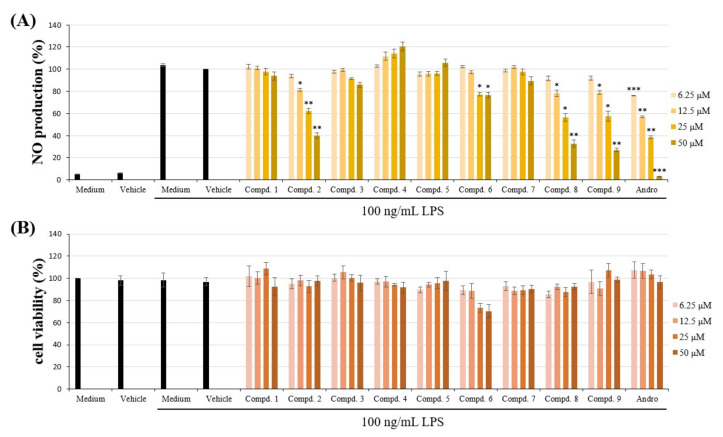
Effects of Compound **1**–**9** from *Hypericum sampsonii* on (**A**) NO production and (**B**) cytotoxicity in LPS-induced RAW264.7 macrophages. Data expressed as means ± SEM *(n =* 3). Asterisks indicate significant differences (* *p* < 0.05, ** *p* < 0.01 and *** *p* < 0.001) from the (media + DMSO + LPS) control group.

**Figure 9 molecules-25-04463-f009:**
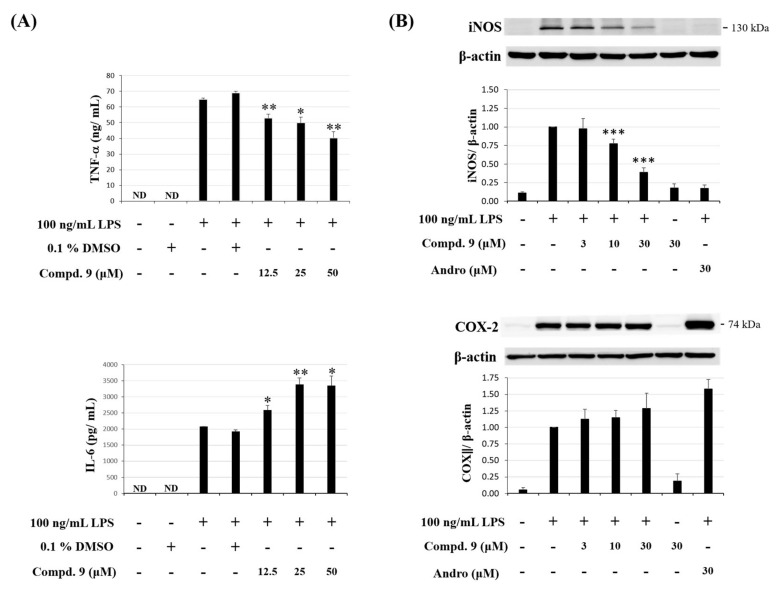
Effect of otogirinin A (**9**) on proinflammatory mediator expression in LPS-induced RAW264.7 macrophages. (**A**) Expression levels of TNF-α and IL-6 determined by ELISA. Data expressed as means ± SEM *(n =* 3). Asterisks indicate significant differences (* *p* < 0.05 and ** *p* < 0.01) from the (media + DMSO + LPS) control group; (**B**) expression of iNOS and COX2 was determined by Western blot analysis. Data expressed as means ± SEM *(n =* 3). Asterisks indicate significant differences (*** *p* < 0.001) from the (media + LPS) control group.

**Figure 10 molecules-25-04463-f010:**
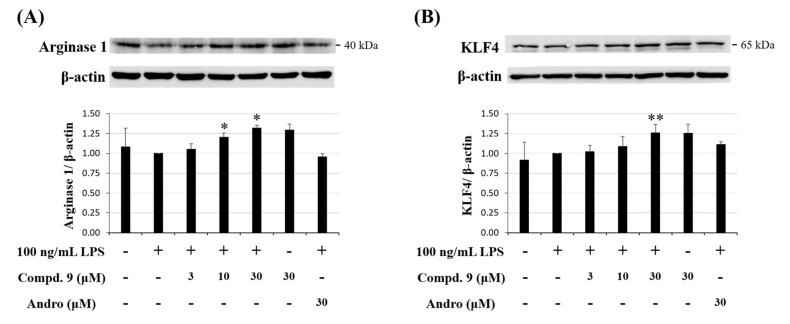
Effect of otogirinin A (**9**) on M2 polarized macrophages in LPS-stimulated RAW264.7 macrophages. (**A**,**B**) Expression of arginase 1 and KLF4 was determined by Western blot analysis. Data expressed as means ± SEM *(n =* 3). Asterisks indicate significant differences (* *p* < 0.05 and ** *p* < 0.01) from the (media + LPS) control group.

**Figure 11 molecules-25-04463-f011:**
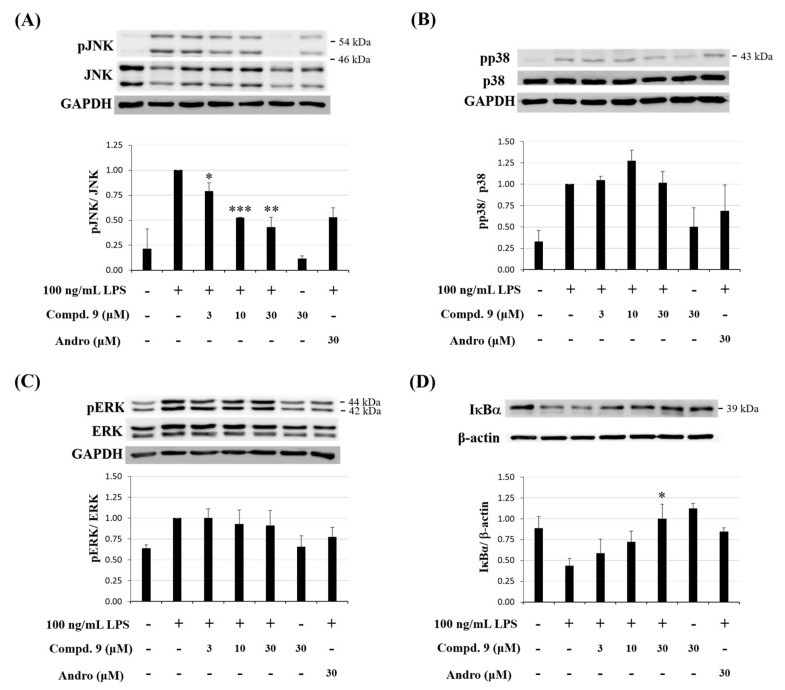
Effect of otogirinin A (**9**) on the phosphorylation of MAPKs and NF-κB pathways in LPS-stimulated RAW264.7 macrophages. (**A**–**C**) Cells pretreated with different concentration of Compound **9** then stimulated with LPS (100 ng/mL). MAPKs and phosphorylation of MAPKs were detected by Western blot antibodies against p-JNK, JNK, p-p38, p38, p-ERK and ERK; (**D**) Cells were pretreated with different concentration of Compound **9** then stimulated with LPS (100 ng/mL). The expression of IκBα was detected by Western blot analysis. Data expressed as means ± SEM *(n =* 3). Asterisks indicate significant differences (* *p* < 0.05, ** *p* < 0.01 and *** *p* < 0.001) from the (media + LPS) control group.

**Figure 12 molecules-25-04463-f012:**
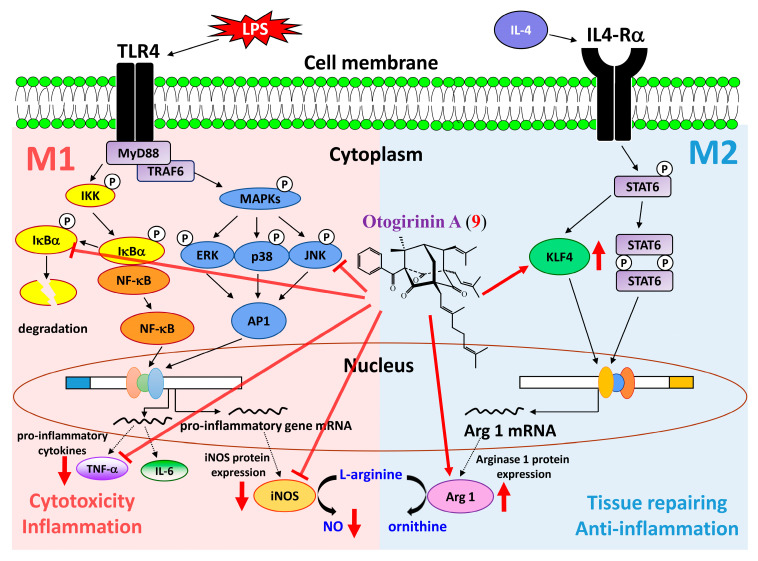
Schematic diagram for anti-inflammatory action of otogirinin A (**9**) in LPS-induced RAW264.7 macrophages.

**Table 1 molecules-25-04463-t001:** Inhibitory effects of Compounds **1**–**9** from the aerial parts of *H. sampsonii* on nitric oxide (NO) generation by RAW264.7 murine macrophages in response to lipopolysaccharide (LPS) ^a^.

Compounds	IC_50_ (μM) ^b^ or (Inh%) ^c^
4-Geranyloxy-2-hydroxy-6-isoprenyloxybenzophenone (**1**)	(6.23 ± 3.71)
Hypericumone A (**2**)	40.32 ± 2.05 **
Hypericumone B (**3**)	(14.12 ± 2.19)
1-Hydroxy-7-methoxyxanthone (**4**) ^d^	elicit NO generation
2-Methoxyxanthone (**5**) ^d^	elicit NO generation
2,4,6-Trihydroxybenzophenone-4-*O*-geranyl ether (**6**)	(23.41 ± 2.56) *
Sampsonin B (**7**)	(10.48 ± 3.52)
Sampsonione J (**8**)	35.25 ± 2.46 **
Otogirinin A (**9**)	32.87 ± 1.60 **
Andrographolide ^e^	19.36 ± 1.72 ***

^a^ Results expressed as average ± SEM *(n =* 3); ^b^ concentration necessary for 50% inhibition (IC_50_). If IC_50_ value of compound was < 50 μM, it is displayed as IC_50_ (μM); ^c^ percentage of inhibition (Inh%) at 50 μM. If IC_50_ value of compound was ≥ 50 μM, it was shown as (Inh%) at 50 μM; ^d^ compound **4** and **5** (50 μM) alone elicit NO generation by LPS-induced murine macrophage cell line RAW264.7.; ^e^ andrographolide was used as positive control; * *p* < 0.05 compared with the control; ** *p* < 0.01 compared with the control; *** *p* < 0.001 compared with the control.

**Table 2 molecules-25-04463-t002:** ^1^H-NMR data for Compounds **1**–**3** (δ in ppm, *J* in Hz).

Position	1 ^a^	2 ^b^	3 ^b^
1			
2			7.99 br d (7.5)
3	6.16 d (2.5)		7.45 br t (7.5)
4			7.55 br t (7.5)
5	5.93 d (2.5)		7.45 br t (7.5)
6		2.08 dd (13.8, 4.1) 2.32 dd (13.8, 2.0)	7.99 br d (7.5)
7		1.80 m	
8			4.35 s
9	7.48 d (7.5)		
10	7.35 br t (7.5)		2.88 m
11	7.43 br t (7.5)		1.14 m 2.20 m
12	7.35 br t (7.5)	7.56 br d (8.4)	2.64 m
13	7.48 d (7.5)	7.33 br dd (8.4, 7.4)	
14		7.45 br t (7.4)	1.98 m 2.32 m
15		7.33 br dd (8.4, 7.4)	5.05 br t (7.6)
16		7.56 br d (8.4)	
17		1.36 s	1.95 m
18		1.44 s	2.02 m
19		1.93 m	5.07 br t (6.8)
20		2.80 br t (10.0)	
21			1.66 s
22		1.85 s	1.58 s
23		5.01 br s 5.25 br s	1.59 s
24			1.67 m 2.15 m
25			5.21 br t (7.4)
26			
27			1.73 s
28			1.60 s
29		2.58 m	0.91 s
30		5.40 br t (7.5)	0.99 s
31			
32		1.69 s	
33		2.11 m	
34		2.14 m	
35		5.10 br t (6.7)	
36			
37		1.62 s	
38		1.68 s	
1ʹ	4.58 d (6.5)		
2ʹ	5.48 br t (6.5)		
3ʹ			
4ʹ	2.11 br t (6.5)		
5ʹ	2.14 m		
6ʹ	5.10 br t (6.5)		
7ʹ			
8ʹ	1.69 s		
9ʹ	1.76 s		
10ʹ	1.62 s		
1ʹʹ	4.18 d (6.0)		
2ʹʹ	4.62 br t (6.0)		
3ʹʹ			
4ʹʹ	1.57 s		
5ʹʹ	1.50 s		
2-OH	12.37 s		
3-OH		3.38 s	

^a^ measured in CDCl_3_ at 500 MHz. ^b^ measured in CDCl_3_ at 600 MHz.

**Table 3 molecules-25-04463-t003:** ^13^C-NMR data for Compounds **1**–**3** (δ in ppm).

Position	1 ^a^	2 ^a^	3 ^a^
1	105.7	80.6	138.5
2	166.1	211.6	128.7
3	94.1	82.6	128.6
4	165.8		133.2
5	92.5	58.6	128.6
6	161.4	36.1	128.7
7	199.6	42.0	196.7
8	142.3	50.8	72.1
9	127.4	212.3	208.4
10	127.5	193.3	47.6
11	130.2	134.5	34.5
12	127.5	129.1	40.3
13	127.4	128.1	43.3
14		132.8	27.4
15		128.1	121.6
16		129.1	136.7
17		21.7	39.8
18		24.5	26.8
19		26.0	124.3
20		41.6	131.3
21		142.5	25.7
22		21.7	16.2
23		118.5	17.7
24			28.1
25			123.6
26			132.4
27			25.8
28			17.8
29		26.3	22.1
30		118.5	26.1
31		139.0	
32		16.2	
33		40.2	
34		26.4	
35		124.1	
36		131.7	
37		17.7	
38		25.8	
1ʹ	65.2		
2ʹ	118.4		
3ʹ	142.3		
4ʹ	39.5		
5ʹ	26.3		
6ʹ	123.6		
7ʹ	131.9		
8ʹ	25.7		
9ʹ	16.7		
10ʹ	17.7		
1ʹʹ	65.1		
2ʹʹ	118.4		
3ʹʹ	137.0.		
4ʹʹ	25.5		
5ʹʹ	18.0		

^a^ measured in CDCl_3_ at 125 MHz.

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
