# Peer review of "Benzophenone and Benzoylphloroglucinol Derivatives from Hypericum sampsonii with Anti-Inflammatory Mechanism of Otogirinin A"

_molecules, 2020, doi:10.3390/molecules25194463_

Round 1
Reviewer 1 Report
The authors address an important knowledge gap in assessing anti-inflammatory activity of compounds isolated from Hypericum sampsonii. The information provided in the manuscript is of utmost significance to enhance the further research on the newly isolated compounds on the developmental strategy of NO production-targeted anti-inflammatory agent. However, the title is misleading as the authors just describe the mechanism of only one derivative of benzoylphloroglucinol (otogirinin A), which has been already known. Moreover, the level of English needs to be improved at several places in the manuscript. Thus, it is my recommendation that this manuscript be accepted after the authors address the above minor and below major comments.
Title should be more defined as the authors have only investigated the anti-inflammatory activity of otogirinin A and not that of compound 8 and 2, which showed reduced NO production.
Introduction needs discussion on the already known compound otogirinin A, sampsoninone J (compound 8).
Results and Discussion:
The authors have only explained the results obtained in the results and discussion section. Results required sufficient discussion in this section or separate the discussion part with the result.
Line 54: why the authors used murine macrophage cell line (RAW264.7) for the study? Not discuss.
Line 175: Why it was obvious that compound 8 and 9 suppressed NO production in RAW264.7 cells? Need further discussion with references.
Line 190: Again need more discussion on the results obtained in this section.
Line 210: Spelling error, change to IL-6.
Line 228: change to “treated”.
Line 233-235: How can the authors say that elevated protein expression of IkBalpha inhibits the translocation of NF-kB, when the authors mentioned that phosphorylation of IkB, activates NF-kB.? Where is NF-kB data? Need clear explanation on this data.
Experimental Sections:
Section 3.4.3 (Line 353) and Section 3.4.5 (Line 371): Why the authors used different exposure conditions for different endpoints. Needs to provide rationale for this different exposure conditions.
Author Response
Comment and suggestions for authors:
The authors address an important knowledge gap in assessing anti-inflammatory activity of compounds isolated from Hypericum sampsonii. The information provided in the manuscript is of utmost significance to enhance the further research on the newly isolated compounds on the developmental strategy of NO production-targeted anti-inflammatory agent. However, the title is misleading as the authors just describe the mechanism of only one derivative of benzoylphloroglucinol (otogirinin A), which has been already known. Moreover, the level of English needs to be improved at several places in the manuscript. Thus, it is my recommendation that this manuscript be accepted after the authors address the above minor and below major comments.
Responses:
Thank you very much for carefully reviewing our manuscript and kindly offering your suggestions. We have revised your remarks as the following statements:
Comment 1:
Title should be more defined as the authors have only investigated the anti-inflammatory activity of otogirinin A and not that of compound 8 and 2, which showed reduced NO production.
Responses:
Title has been corrected, in accordance with comments.
The title has been corrected as “Benzophenone and benzoylphloroglucinol derivatives from Hypericum sampsonii with anti-inflammatory mechanism of otogirinin A”.
Comment 2:
Introduction needs discussion on the already known compound otogirinin A (9), sampsoninone J (compound 8).
Responses:
More discussions of otogirinin A (9) and sampsoninone J (8) have been added in “Introduction” as follows:
“Sampsoninone J (8) and otogirinin A (9) belong to the polyprenylated benzoylphloroglucinol derivatives with an unusual adamantyl skeleton. According to past studies, there was no biological activity report for 8 and 9, except that 8 showed no significant cytotoxicity againt P338 cell line [18]. However, their analogous benzoylphloroglucinol derivatives, garcimultiflorone G [19] and sampsonione B [20] had been reported to exhibit anti-inflammatory activity.”.
Results and Discussion:
Comment 3:
The authors have only explained the results obtained in the results and discussion section. Results required sufficient discussion in this section or separate the discussion part with the result.
Responses:
More discussions have been added in every sections (2.3.2‒2.3.5), in accordance with the reviewer’s comments.
Comment 4:
Line 54: why the authors used murine macrophage cell line (RAW264.7) for the study? Not discuss.
Responses:
The description “The murine macrophage cell line, RAW264.7, is often used to initially screen natural products for anti-inflammatory activity [21,22].” has been added, in accordance with the reviewer’s comments.
Comment 5:
Line 175: Why it was obvious that compound 8 and 9 suppressed NO production in RAW264.7 cells? Need further discussion with references.
Responses:
More discussions have been added as follows:
Nitric oxide (NO) is a mediator in the inflammatory response involved in host defense. The anti-inflammatory effects of the compounds isolated from the aerial parts of H. sampsonii were evaluated by suppressing lipopolysaccharide (LPS)-induced NO generation in murine macrophage cell line RAW264.7. The inhibitory activity data of the isolated compounds 1–9 on NO generation by macrophages are shown in Table 1. The anti-inflammatory agent, andrographolide (Andro), was used as positive control, which had been reported to exhibit gastroprotective and ulcer preventive effects, along with its well-documented anti-inflammatory effects [30]. The results of anti-inflammatory assays show that sampsonione J (8) and otogirinin A (9) obviously suppressed NO production comparable with andrographolide, in a concentration-dependent manner in RAW264.7 macrophages (Figure 8A and Table 1). Compounds 8 and 9 are benzoylphloroglucinol derivatives with inhibition (%) values of 67.20 ± 3.29 and 73.10 ± 1.59 %, respectively, against LPS-induced NO production at 50 μM. Otogirinin A (9) is the most effective among the isolated compounds, with IC50 = 32.87 ± 1.60 μM, against LPS-induced NO generation. Both of the compounds 8 and 9 did not show significant cytotoxicity against RAW264.7 macrophages (Figure 8B), which suggested that inhibitory activities of compounds 8 and 9 on LPS-induced NO production did not resulted from their cytotoxicities.
Reference:
[30] Saranya, P.; Geetha, A.; Selvamathy, S.N. A biochemical study on the gastroprotective effect of andrographolide in rats induced with gastric ulcer. Indian J. Pharm. Sci. 2011, 73, 550–557.
Comment 6:
Line 190: Again need more discussion on the results obtained in this section.
Responses:
More discussion on the results has been added, in accordance with the reviewer’s comments.
Comment 7:
Line 210: Spelling error, change to IL-6.
Responses:
“κL-6” have been corrected as “IL-6”, in accordance with the reviewer’s comments.
Comment 8:
Line 228: change to “treated”.
Responses:
“treating” have been corrected as “treated”, in accordance with the reviewer’s comments.
Comment 9:
Line 233-235: How can the authors say that elevated protein expression of IkBalpha inhibits the translocation of NF-κB, when the authors mentioned that phosphorylation of IκB, activates NF-κB? Where is NF-κB data? Need clear explanation on this data.
Responses:
More discussions about “IκB” and “NF-κB” have been added as “NF-κB is also potential target in the inflammation process and known to be a molecular target of anti-inflammatory drugs. Upon LPS activation, IκB kinase (IKK) phosphorylates IκBα and triggers ubiquitin-dependent IκBα degradation in the proteasome, resulting in rapid and transient nuclear translocation of NF-κB [37]. The degradation of IκBα correlated with NF-κB activation. Therefore, we examined if otogirinin A (9) could suppressed the degradation of IκBα, and translocation of NF-κB into the nucleus. The result indicated that otogirinin A (9) elevated the protein expression of IκBα, which attenuated IκBα degradation and thus inhibited the translocation of NF-κB into the nucleus in LPS-stimulated RAW264.7 macrophages (Figure 11D). These results suggested that suppression of degradation of IκBα may be involved in the inhibitory effect of otogirinin A (9) on LPS-stimulated NF-κB activation in RAW264.7 cells.”
And NF-κB data was shown in perspective of the amount of IκBα (Figure 11D). As we mentioned above, LPS stimulation induced the degradation of IκBα and activation of NF-κB in RAW264.7 cells. The result indicated that otogirinin A (9) elevated the protein expression of IκBα, which attenuated IκBα degradation and thus inhibited the translocation of NF-κB into the nucleus in LPS-stimulated RAW264.7 macrophages.
Experimental Sections:
Comment 10:
Section 3.4.3 (Line 353) and Section 3.4.5 (Line 371): Why the authors used different exposure conditions for different endpoints. Needs to provide rationale for this different exposure conditions.
Responses:
In the experimental sections, section 3.4.3 and section 3.4.5 used the same procedure. Both of the experiments were pre-treated with compounds (0, 6.25, 12.5, 25, and 50 μM) or vehicle (0.1% DMSO) for 1 hour, respectively, in the presence of LPS (100 ng/mL) and incubated under the same condition for 20 h. The cultured cells were then centrifuged, and the supernatants were used for NO-production, TNF-α, and IL-6 measurement. So we use the same exposure conditions for different endpoints.

Reviewer 2 Report
I have reviewed an article: "Benzophenone and benzoylphloroglucinol
derivatives from Hypericum sampsonii with anti inflammatory activity". Authors have detected three new compounds in hexane extracts and determined their structures. They have determined their cytotoxic and antiinflammatory properties and proposed pathways responsible for these effects. Article is well written, all conclusions are made from experimental data. Discussion section is well written so I think this paper can be accepted for publication in its present form.
Author Response
Thank you very much for carefully reviewing our manuscript and kindly thinking this paper can be accepted for publication in its present form.

Reviewer 3 Report
This manuscript clearly proved the key compound for anti-inflammatory activity from Hypericum sampsonii with underlying action mechanisms. Experiment was performed logically and manuscript was well-organized.
No major points was found and only correction on typos are required for the publication in Molecules.
Author Response
Some typos were checked again, in accordance with the reviewer’s comments.
Reviewer 4 Report
Please check and revise.
-------------------------------------------------------------------------------------------------------------------------
Compounds 1-3 structural analysis: For 1, it is unnecessary that ROESY and NOESY analysis. Cpd 1 is no chiral in the structure. For 2 and 3, NOESY analysis is unnecessary. ROESY and NOESY analyses are equivalent analyses.
Table 1.: Why didn’t you unify the results to IC50?
-------------------------------------------------------------------------------------------------------------------------
Author Response
Comment and suggestions for authors:
Please check and revise.
Responses:
Thank you very much for carefully reviewing our manuscript and kindly offering your suggestions. We have explained your remarks as the following statements:
Comment 1:
Compounds 1‒3 structural analysis: For 1, it is unnecessary that ROESY and NOESY analysis. Cpd 1 is no chiral in the structure. For 2 and 3, NOESY analysis is unnecessary. ROESY and NOESY analyses are equivalent analyses.
Responses:
For compound 1, the position of isoprenyloxy and geranyloxy groups should be verified by NOESY analysis. For compound 2 and 3, we also need to verify relative stereochemistry of substituents such as isoprenyl and geranyl groups… by ROESY.
Comment 2:
Table 1.: Why didn’t you unify the results to IC50?
Responses:
As we mentioned in the Table 1, if IC50 value of compound was < 50 μM, it was displayed as IC50 [μM]. If IC50 value of compound was ≥ 50 μM, it was shown as (Inh%) at 50 μM because the inhibition effect (IC50 value ≥ 50 μM) of tested compound on NO generation was not significant. Therefore, we presented our results in this form.
Round 2
Reviewer 1 Report
The authors have addressed to all comments raised in a previous round of review. The overall scientific content in the manuscript is now well documented, which further improves the comprehension of text.